# The JAK1/2 Inhibitor Baricitinib Mitigates the Spike-Induced Inflammatory Response of Immune and Endothelial Cells In Vitro

**DOI:** 10.3390/biomedicines10092324

**Published:** 2022-09-19

**Authors:** Amelia Barilli, Rossana Visigalli, Francesca Ferrari, Giulia Recchia Luciani, Maurizio Soli, Valeria Dall’Asta, Bianca Maria Rotoli

**Affiliations:** 1Laboratory of General Pathology, Department of Medicine and Surgery, University of Parma, 43125 Parma, Italy; 2Immunohematology and Transfusion Medicine, University Hospital of Parma, 43125 Parma, Italy

**Keywords:** baricitinib, chemokines, COVID-19, endothelium, immune innate cells, JAK-STAT, IL-8

## Abstract

The purpose of this study was to examine the effect of the JAK-STAT inhibitor baricitinib on the inflammatory response of human monocyte-derived macrophages (MDM) and endothelial cells upon exposure to the spike S1 protein from SARS-CoV-2. The effect of the drug has been evaluated on the release of cytokines and chemokines from spike-treated MDM, as well as on the activation of endothelial cells (HUVECs) after exposure to conditioned medium collected from spike-activated MDM. Results obtained indicate that, in MDM, baricitinib prevents the S1-dependent phosphorylation of STAT1 and STAT3, along with the induction of IP-10- and MCP-1 secretion; the release of IL-6 and TNFα is also reduced, while all other mediators tested (IL-1β, IL-8, RANTES, MIP-1α and MIP-1β) are not modified. Baricitinib is, instead, poorly effective on endothelial activation when HUVECs are exposed to supernatants from S1-activated macrophages; the induction of VCAM-1, indeed, is not affected by the drug, while that of ICAM-1 is only poorly inhibited. The drug, however, also exerts protective effects on the endothelium by limiting the expression of pro-inflammatory mediators, specifically IL-6, RANTES and IP-10. No effect of baricitinib has been observed on IL-8 synthesis and, consistently, on neutrophils chemiotaxis. Our in vitro findings reveal that the efficacy of baricitinib is limited, with effects mainly focused on the inhibition of the IL-6-mediated inflammatory loop.

## 1. Introduction

Baricitinib, a selective and reversible inhibitor of the Janus kinases JAK1 and JAK2, previously approved for the treatment of rheumatoid arthritis [1], has been recently licensed for the treatment of severe Coronavirus disease 2019 (COVID-19) (COVID-19 Treatment Guidelines Panel. Coronavirus Disease 2019 (COVID-19) Treatment Guidelines. National Institutes of Health. Available at https://www.covid19treatmentguidelines.nih.gov, accessed on 19 July 2022). COVID-19, caused by SARS-CoV-2 infection, is characterized by a profound dysfunction of the upper and lower respiratory tract, with severity ranging from mild to moderate respiratory failure, to acute respiratory distress syndrome (ARDS) [2]. It is widely recognized that, in COVID-19, cell damage is caused not only directly by the virus, but also by the host immune response unleashed by the infection, whose magnitude deeply influences the severity of the disease. Indeed, a massive overproduction of cytokines (the so called “cytokine storm”) is now considered one of the major causes of clinical worsening toward ARDS and multiple-organ failure in the later phases of the disease [3,4,5,6,7]. In addition to this concern, elevated serum levels of cytokines and chemokines have been described in patients with COVID-19, where they correlate with the severity of the disease [8,9]. Circulating levels of IL-6, in particular, have been shown to be significantly higher in severe than mild to moderate disease and to predict COVID-19 severity and survival [10,11]. Although the mechanisms by which SARS-CoV-2 infection induces cytokine overproduction are not yet completely understood, it is evident that therapeutical approaches targeting cytokine responses, besides anti-viral drugs, deserve particular attention to decrease morbidity and mortality in COVID-19 patients [12]. In this context, many agents able to inhibit one or more components of the proinflammatory cascade have been recently investigated in clinical trials, so as to improve the outcome of severe patients [13,14]. Among them, the monoclonal antibodies targeting the IL-6 receptor, tocilizumab and sarilumab, proved effective in improving the clinical conditions of severe patients [15,16,17,18]. Moreover, alternative therapeutic strategies consist of agents targeting the various molecular pathways driving the secretion of inflammatory mediators (see [19] for review). In particular, US Food and Drug Administration (FDA)-approved ponatinib, a kinase inhibitor, is able to inhibit cytokine release caused by the N-terminal domain (NTD) of the SARS-CoV-2 spike protein [20]. Similarly, inhibitors of JAK-STAT represent an encouraging therapeutic strategy to counteract the progression to more severe conditions [21,22], given the deep involvement of this transduction pathway in the onset of cytokine storms in COVID-19 [23]. Baricitinib, in particular, has been recently approved by FDA for the treatment of hospitalized patients requiring supplemental oxygen [24]; in clinical trials, baricitinib treatment was associated with clinical and radiologic recovery, a rapid decline in SARS-CoV-2 viral load, and inflammatory markers including IL-6 levels [25,26,27,28,29].

The direct effect of this drug on the inflammatory responses of human cells in vitro remains, however, to be elucidated. Thus far, only one study has examined the effect of the exogenous addition of baricitinib on the immune-specific viral response; more precisely, by using a whole-blood platform of COVID-19 and “NO COVID-19″ individuals, the authors demonstrate that the inhibitor shuts down several cytokines in response to SARS-CoV-2 peptides [30].

In this context, the aim of the present contribution has been to explore the impact of baricitinib on the spike-dependent induction of the inflammatory response in vitro. To this end, the effect of the drug has been evaluated on cytokine release by S1-treated monocyte-derived macrophages (MDM) as well as on the inflammatory profile induced in endothelial cells by the exposure to the MDM-derived pool of cytokines.

## 2. Materials and Methods

### 2.1. Cell Models

Monocytes were isolated from buffy coats obtained from eight normal healthy donors provided by Immunohematology and Transfusion Medicine, University Hospital of Parma, Italy. Buffy coats, diluted 1:4 with PBS, were layered on 15 mL Lympholyte H (Euroclone, Milano, Italy) and centrifuged at 800× *g* for 20 min at 20 °C. PBMCs at the interface were collected and, after two washes in PBS, were suspended in complete growth medium (RPMI containing 10% endotoxin-free FBS) and seeded on plasticware. After a 30-min incubation at 37 °C, nonadherent cells were removed with three vigorous washes in PBS. Monocyte derived macrophages (MDMs) were obtained by incubating monocytes in complete growth medium added with 50 ng/mL of recombinant human Granulocyte Mϕ-Colony-Stimulating Factor (GM-CSF, Vinci-Biochem, Firenze, Italy) for up to 6 d. The study was approved by the local ethical committee (460/2021/TESS/UNIPR).

Human Umbilical Vein Endothelial Cells (HUVEC) were purchased from Cell Applications, Inc. (Merck, Milano, Italy) and routinely grown according to the manufacturer’s instructions.

### 2.2. Experimental Treatments

Monocyte-derived macrophages (MDM) were incubated in the absence and in the presence of 5 nM S1 subunit of SARS-CoV-2 spike recombinant protein (ARG70218; Arigo Biolaboratories, Taiwan). To exclude any possible contamination by lipopolysaccharide (LPS), S1 was pre-mixed with 2 µg/mL of the LPS inhibitor Polymyxin B (Merck) and incubated for 30 min at RT before the addition to the cell cultures. MDM were then incubated for different times in the absence (control) or presence of S1 spike (S1); when required, 1 µM baricitinib was added 1 h before the incubation with S1 (S1+baricitinib). The culture media obtained under these experimental conditions were collected and pooled as conditioned medium from control MDM (CM_cont), S1-treated MDM (CM_S1) and S1+baricitinib-treated MDM (CM_S1_baricitinib); these CM were then quantified for cytokine content (see below) and employed to test the effects on leukocytes migration (see below). CM_cont and CM_S1 were also employed for the treatment of HUVEC cells; where indicated, these cells were pre-treated with 1 µM baricitinib for 1 h before the addition of CM_S1 and the inhibitor was then left in the culture medium throughout the experiment.

### 2.3. RT-qPCR Analysis

The analysis of gene expression was performed with RT-qPCR, as already described [31]. Briefly, 1 µg of total RNA was reverse transcribed with a RevertAid RT Reverse Transcription kit (Thermo Fisher Scientific, Monza, Italy) and 20 ng of cDNA underwent qPCR on a StepOnePlus Real-Time PCR System (Thermo Fisher Scientific). Forward/reverse primer pairs detailed in Table 1 were employed for qPCR analysis along with TaqMan™ Fast Advanced Master Mix or PowerUp Sybr™ Green Master Mix (Thermo Fisher Scientific); the expression of the genes of interest was calculated relatively to that of the housekeeping gene (*RPL15*) with either the ∆∆Ct or the 2^∆Ct^ method [32], as specified in each Figure’s legend.

### 2.4. Cytokine Analysis

Cell culture supernatants from treated MDM were collected, centrifuged at 300× *g* for 10 min to remove particulates, and stored at −20 °C. For the analysis, Human Luminex Discovery Assay (R&D Systems, Bio-techne, Milano, Italy) was employed for the detection of CCL2/MCP-1, CCL3/MIP-1α, CCL4/MIP-1β, CCL5/RANTES, CXCL8/IL-8, CXCL10/IP-10, IL-1β, IL-6 and TNFα, according to the manufacturer’s instructions. Given the expected variability in cytokine content, different dilutions for each sample (1:10, 1:100 and 1:10,000 in RD6-52 calibrator diluent) were employed so as to fall within the range of concentration of standard curves for each analyte. The concentrations of cytokines are given as ng/mL.

### 2.5. Western Blot Analysis

The analysis of protein expression was performed on cell lysates obtained with LDS sample buffer (Thermo Fisher Scientific), as already described [33]. 20 µg of proteins were separated on Bolt™ 4–12% Bis-Tris mini protein gel (Thermo Fisher Scientific) and transferred to PVDF membranes (Immobilon-P membrane, Thermo Fisher Scientific). Membranes were incubated for 1 h at RT in a blocking buffer (Tris-buffered saline solution—TBS; 50 mM Tris-HCl pH 7.5, 150 mM NaCl) added with 3% non-fat dried milk and then overnight at 4 °C with primary antibodies in TBST (TBS + 0.5% Tween) containing 5% BSA. The following rabbit polyclonal antibodies (1:2000, Cell Signaling Technology, Euroclone, Milano, Italy) were employed: anti-phospho-STAT1 (Tyr701), anti phospho-STAT3 (Tyr705), anti-phospho-NF-κB p65 (Ser536), anti phospho-IκBα (Ser32/36), anti-ICAM-1, or anti-VCAM-1. Anti-vinculin mouse monoclonal antibody (1:2000, Merck) was used as loading control. Horseradish peroxidase (HRP)-conjugated secondary antibodies (anti-rabbit and anti-mouse IgG) were provided by Cell Signaling Technology (1:10,000). Immunoreactivity was visualized with SuperSignal™ West Pico PLUS Chemiluminescent HRP Substrate (Thermo Fisher Scientific). Western Blot images were captured with an iBright FL1500 Imaging System (Thermo Fisher Scientific) and analysed with iBright Analysis Software.

### 2.6. Leukocytes Transendothelial Migration

Human polymorphonucleated (PMN) and CD3+ T lymphocytes were isolated by means of Lympholyte^®^-poly (Euroclone, Milano, Italy) density gradient. After centrifugation, PMN in the lower band and mononuclear cells in the upper band were collected, washed twice with RPMI1640 medium and resuspended in the same medium. CD3+ T cells were then isolated from mononuclear cells through magnetic separation with CD3 MicroBeads (Miltenyi Biotec, Bologna, Italy). PMN and CD3+ cells migration toward conditioned media from MDM was then evaluated with the QCMTM Chemiotaxis (3 µM) 24-well Cell Migration Assay (Millipore, Merk, Milano, Italy). Briefly, transwell inserts were coated with collagen and HUVECs (30 × 10^3^ cells/insert) were seeded into the upper compartment. After 48 h, 5 × 10^5^ PMN or 4 × 10^5^ T lymphocytes were placed into the upper chamber of the insert and were allowed to transmigrate through the endothelial layer towards the lower compartment containing CM_cont, CM_S1 and CM_S1_baricitinib. After 4 or 24 h, the amount of migrated PMN or T cells, respectively, was determined as specified in the assay protocol.

### 2.7. Statistical Analysis

GraphPad Prism 9 (GraphPad Software, San Diego, CA, USA) was used for statistical analysis. *p* values were calculated with one-way ANOVA for matched measures and the Holm-Sidak correction for multiple comparisons or with Student *t* test for paired data, as specified in the legend of each Figure. *p* values < 0.05 were considered to be statistically significant.

### 2.8. Materials

Endotoxin-free fetal bovine serum (South America origin; EU Approved) was purchased from Euroclone (Milano, Italy); Merck (Milano, Italy) was the source of baricitinib, as well as of all of the other chemicals, unless otherwise specified.

## 3. Results

In order to investigate the efficacy of baricitinib in the modulation of the inflammatory phenotype of human macrophages, we first tested the effect of the drug on the expression and release of cytokines and chemokines induced by SARS-CoV-2 spike S1 protein in monocytes-derived macrophages (MDM). In a recent contribution we demonstrated, that MDM respond to the viral protein by secreting massive amounts of inflammatory mediators [34]. Here, by further addressing this issue, we show that, despite the high variability of cell responses due to the use of monocytes from different donors, the incubation with S1 for 4 h significantly induces the expression of IL-1β, TNFα and IL-6, as well as of the chemokines CXCL8/IL-8, CCL2/MCP-1, CXCl10/IP-10, CCL5/RANTES, CCL3/MIP-1α and CCL4/MIP-1β (Figure 1). In agreement with the anti-inflammatory potential of baricitinib, the addition of the drug to the incubation medium partially, but significantly, modifies the expression of these mediators, lowering the induction of IL-6 and TNFα, and completely preventing that of MCP-1 and IP-10 (Figure 1).

A similar pattern is observed when addressing the amount of cytokines released in supernatants by MDM in response to a 24 h-stimulation with S1 protein in the absence or in the presence of baricitinib (Figure 2). In line with what was observed at the mRNA level, the secretion of all the inflammatory mediators tested is also markedly stimulated by the spike protein; moreover, while the release of IL-6 and TNFα is modestly but significantly reduced by the JAK/STAT inhibitor, that of IP-10 and MCP-1 is completely suppressed, whereas the amount of all the other compounds measured (i.e., IL-1β, IL-8, RANTES, MIP-1α and MIP-1β) remains unsensible to the presence of baricitinib. This latter finding is particularly intriguing when considering IL-8, whose secretion is impressively stimulated by S1 protein (with an increase from 4.1 + 0.5 ng/mL under control conditions to 165.8 ± 41.4 ng/mL in spike-treated cells), but completely unaffected, or even slightly stimulated, by the presence of the JAK/STAT inhibitor.

The incubation media of MDM maintained under the different experimental conditions (control, S1, and S1+baricitinib) were then collected and employed as conditioned media (CM_cont, CM_S1, and CM_S1_baricitinib, respectively) to test their chemoattractant potential on human leucocytes with a migration assay. The results obtained clearly indicate that CM_S1 significantly stimulates the transendothelial migration of neutrophils with respect to CM_cont, and that the presence of baricitinib (CM_S1_baricitinib) does not prevent this effect (Figure 3, Panel A). The same assay was also performed by employing CD3+ T cells. As shown in Panel B, CM_S1 evoked an increase in the transendothelial migration of human T lymphocytes, with baricitinib promoting a modest but significant decrease of the chemiotaxis of these cells.

To explore the molecular pathways underlying the inflammatory profile induced by the S1 protein in MDM, the effect of S1 was investigated, both in the absence and in the presence of baricitinib, on the activation of STATs and NF-ĸB transcription factors. As shown in Figure 4, the incubation of MDM with S1 alone causes the phosphorylation of both STAT1 and STAT3 after 90 min; the effect is transitory, with a peak after 4 h and a decline after 24 h. As expected, baricitinib completely prevents the phosphorylation of both STAT1 and STAT3, at any time. The activation of NF-ĸB is more rapid, since the phosphorylation of p65 subunit, as well as that of the inhibitor IĸBα, are already detectable after 30 min, last for 4 h and decrease at basal levels after 24 h of incubation. Baricitinib is only modestly effective on the activation of NF-ĸB, with a partial inhibition of both p-p65 and p-IĸBα observed after 90 min, 4 h and 24 h.

The efficacy of baricitinib has been evaluated on the modulation of the response of endothelium to inflammatory mediators produced by spike-treated macrophages. To this end, HUVEC cells have been treated with conditioned medium obtained from MDM incubated for 24 h in the absence (CM_cont) or in the presence of S1 (CM_S1); under this latter condition, the effect of baricitinib has been tested by adding the inhibitor to the cells 1 h before the incubation with CM_S1 (Figure 5). As already reported [34], a clear-cut endothelial activation occurs upon incubation with CM_S1, as demonstrated by the appearance of adhesion molecules Intracellular Cell Adhesion Molecule-1 (ICAM-1) and Vascular Cell Adhesion Molecule-1 (VCAM-1), while the addition of baricitinib proves completely ineffective on the expression of VCAM-1, and a modest reduction of ICAM-1 is observed at both the mRNA and protein level.

In the same cells, the incubation with CM_S1 also massively stimulates the expression of many cytokines and chemokines, namely IL-1β, TNFα, IL-6, IL-8, MCP-1, IP-10 and RANTES (Figure 6). The addition of baricitinib completely prevents the induction of IL-6, IP-10 and RANTES, while leaving unaffected the expression of all other mediators tested.

## 4. Discussion

The purpose of this study was to investigate the efficacy of baricitinib, an anti-inflammatory drug that selectively inhibits JAK1/2 signalling, on the inflammatory status induced by spike S1 protein in human innate immune cells, as well as in endothelial cells. To this end, the effect of the drug has been evaluated on the production of cytokines by S1-treated macrophages, as well as on the inflammatory response elicited in endothelial cells (HUVEC) by the mixture of cytokines and chemokines released by spike-activated macrophages.

The JAK/STAT pathway is currently recognized as one of the signalling mechanisms mainly involved in the induction of cytokines and chemokines in COVID-19 [19,35]. Thus, targeting JAK1/2 represents a valid therapeutic strategy for the treatment of the disease. Among the JAK-STAT inhibitors, baricitinib has received FDA’s authorization for the treatment of patients with severe COVID-19, since it proved effective in preventing the progression to a severe form of the disease and resulted in reduced hospitalization and mortality [29,36,37,38]. Besides the anti-inflammatory properties, this drug is endowed with potential anti-viral effects mediated by the inhibition of Numb-associated kinases [25]. On the other hand, the inhibition of JAK1/2 has been postulated to produce an impairment of interferon-mediated antiviral response, with a potential facilitating effect on the evolution of SARS-CoV-2 infection [39], and to an increased susceptibility to other viral infections, as described in patients with rheumatoid arthritis [40].

Here, we demonstrate that the stimulation of monocyte-derived macrophages (MDM) with S1 protein induces STAT1 and STAT3 phosphorylation, as well as the activation of NF-ĸB transcription factor. As expected, baricitinib completely prevents the activation of STATs and is effective in inhibiting the release of cytokines and chemokines from spike-activated macrophages, thus sustaining a direct involvement of these transcription factors in the release of at least some inflammatory mediators in COVID-19.

Recently, hallmarks predisposing to severe illness have been identified that describe the main alterations occurring in the course of the disease [41]; among them, high circulating levels of the cytokine IL-6 and the chemokines MCP-1 and IP-10 have been identified as reliable markers to predict COVID-19 severity and survival [42,43,44,45]. The efficacy of baricitinib in inhibiting the secretion of these mediators has already been reported in synovial fibroblasts from patients with rheumatoid arthritis [46], where the drug was initially employed. In our hands, a similar pattern of inhibition has been observed on spike-treated MDM, where the release of IL-6 is significantly reduced by the addition of baricitinib, while that of MCP-1 and IP-10 is even completely prevented.

According to our results, however, the beneficial effects of the drug are limited to only some of the inflammatory mediators stimulated by Sars-CoV-2 spike. The secretion of IL-8, the most abundant cytokine detected in the culture medium of S1-treated macrophages, remains, for instance, completely unaffected by the addition of baricitinib. Similar results have been shown in vivo by Bronte et al., who reported a tendency of IL-8 plasma levels to increase after 7 d of treatment with baricitinib, despite an evident drop of IL-6 during the same time [29]. IL-8 expression in coronavirus infected cells has been reported under AP-1 control [47], and, similarly, we found the same transcription factor involved in IL-8 synthesis in epithelial A549 cells [48]. Given the primary role of this chemokine as a potent attractant for neutrophils [49,50], the inefficacy of baricitinib on IL-8 release could explain why the transendothelial migration of neutrophils in vitro is stimulated by the CM_S1 medium and unaffected by baricitinib. On the contrary, the efficacy of the drug in limiting the migration of T lymphocytes likely comes from the clearance in CM_S1 conditioned medium of IP-10 and MCP-1 chemokines, known chemoattractants for mononuclear cells [51,52].

We have recently demonstrated that the spike S1 protein has negligible effects on the inflammatory profile of endothelial and alveolar epithelial cells; these cells, however, display an evident inflammatory response when exposed to conditioned medium obtained from spike-activated macrophages, as evidenced by the release of cytokines and chemokines and by the induction of endothelial adhesion molecules [34,48]. In this study, we show that the addition of baricitinib selectively prevents the induction of IL-6 in HUVEC, RANTES and IP-10, but has no effect on all other mediators tested. Among the cytokines, IL-6, whose level in the plasma of COVID-19 patients is very high, is particularly important for predicting the disease progression and severity [10,53]. This cytokine is known to stimulate the production of cytokines and chemokines by endothelial and epithelial cells through the induction of STAT3, resulting in the amplification of the inflammatory cascade. In this context, our findings, showing that baricitinib completely prevents the induction of IL-6 expression in endothelial cells, imply that the drug can play a central role in the modulation of the inflammatory response by interrupting the IL-6-based positive loop. On the contrary, the inefficacy of baricitinib on the expression of TNFα under the same experimental conditions indicates that the induction of these mediators by the pool of cytokines present in MDM conditioned medium is independent from STATs. The drug is, however, effective in preventing the small increase induced in endothelial cells by spike alone [54], confirming that S1-induced TNFα expression is under the JAK/STAT pathway in these cells as well.

The increase of circulating cytokines and chemokines during Sars-CoV-2 infection is known to cause the synthesis of adhesion molecules, such as ICAM-1 and VCAM-1 [34,55,56,57]. This induction is mainly modulated by TNF-α through the activation of the NF-κB pathway [58], although IL-6 has also been shown to regulate these adhesion molecules through the activation of STAT3 [59,60]. In line with these findings, a recent contribution by Meyer et al. showed that tocilizumab, a monoclonal antibody targeting the IL-6 receptor, is effective on VCAM-1 and ICAM-1 in TMNK-1 and EAhy926 endothelial cell lines [61]. In our hands, the activation of HUVEC by CM from S1-activated macrophages is poorly affected by the addition of baricitinib, since the inhibitor is completely ineffective on VCAM-1 induction and only partially active on ICAM-1. Overall, our results suggest that baricitinib exerts only a partial protective effect on COVID-19-associated endothelial dysfunction.

## 5. Conclusions

Our findings in vitro highlight some potential clinical implications of the use of baricitinib for the treatment of COVID-19. One consideration comes from the efficacy of the drug on IL-6-mediated effects, which makes the drug beneficial in limiting the inflammatory loop between monocytes and endothelial cells. On the other hand, however, the inefficacy on the synthesis of IL-8, as well as on the induction of adhesion molecules in endothelial cells, implies that the therapy with baricitinib in COVID-19 could be not able to prevent the excessive enrolment of granulocytes and their deleterious effect on tissues.

## Figures and Tables

**Figure 1 biomedicines-10-02324-f001:**
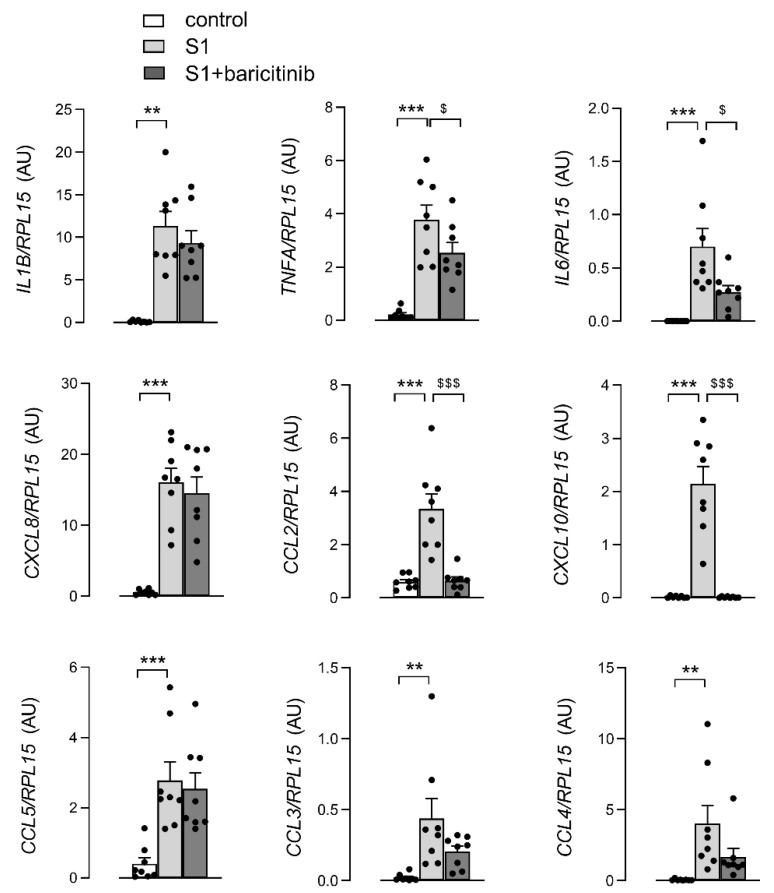
Monocyte-derived macrophages (MDM) were left untreated (control) or incubated for 4 h with 5 nM spike S1 protein in the absence (S1) or in the presence of 1 µM baricitinib (S1+baricitinib). The expression of the indicated genes was measured by means of RT-qPCR and calculated relatively to that of the housekeeping gene *RPL15*, by employing the ∆∆Ct method. Bars are means ± SEM of eight independent determinations (single dots), each performed in duplicate. ** *p* < 0.01, *** *p* < 0.001 vs. untreated cells (control); ^$^
*p* < 0.05, ^$$$^
*p* < 0.001 vs. S1-treated cells with one-way ANOVA.

**Figure 2 biomedicines-10-02324-f002:**
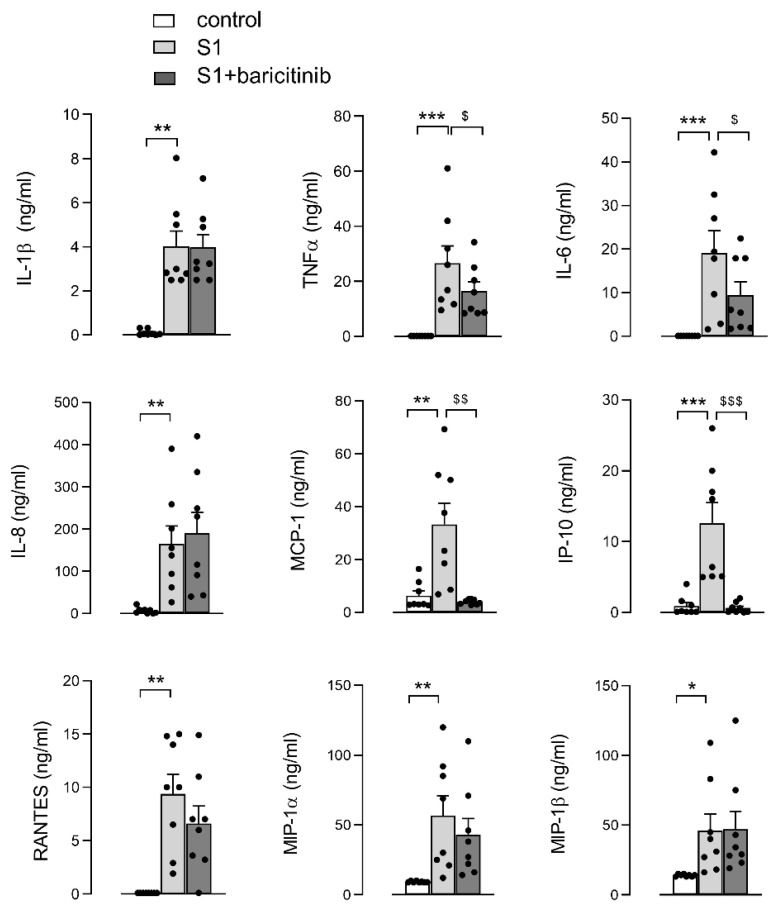
MDM were left untreated (control) or incubated for 24 h with 5 nM spike S1 protein in the absence (S1) or in the presence of 1 µM baricitinib (S1+baricitinib). Cytokines released in the medium were quantified with an ELISA assay, as described in Materials and Methods. Bars are means ± SEM of eight independent experiments (single dots). * *p* < 0.05, ** *p* < 0.01, *** *p* < 0.001 vs. untreated cells (control); ^$^
*p* < 0.05, ^$$^
*p* < 0.01, ^$$$^
*p* < 0.001 vs. S1-treated cells with one-way ANOVA.

**Figure 3 biomedicines-10-02324-f003:**
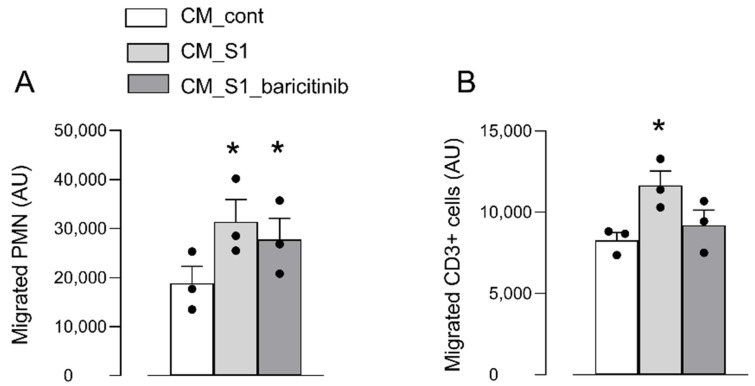
Human polymorphonucleates (PMN, Panel (**A**)) and T lymphocytes (CD3+ cells, Panel (**B**)) were isolated and allowed to migrate through confluent HUVECs grown on transwell inserts towards CM_cont, CM_S1 or CM_S1_baricitinb. After 4 h (PMN) or 24 h (CD3+ cells), the amount of transmigrated cells in the lower compartment was determined as described in Materials and Methods. Bars are the mean ± SEM of three independent determinations (single dots). * *p* < 0.05 with one-way ANOVA.

**Figure 4 biomedicines-10-02324-f004:**
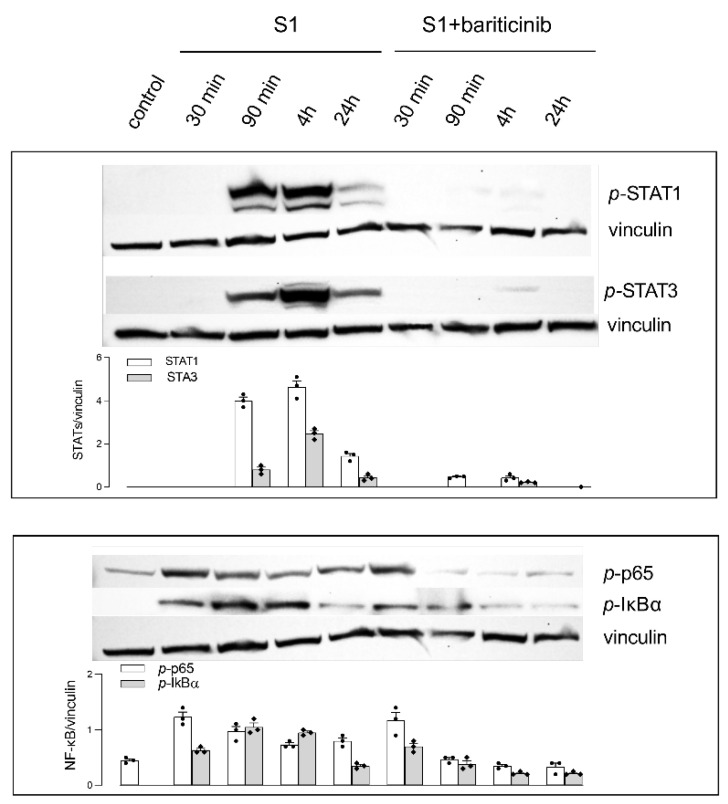
Monocyte-derived macrophages (MDM) were left untreated (control) or incubated for the indicated times with 5 nM spike S1 in the absence (S1) or in the presence of 1 µM baricitinib (S1+baricitinib). The expression of the indicated proteins was assessed by means of western blot analysis, as detailed in Materials and Methods; representative blots are shown, along with the mean of the densitometric analyses (bars) of three different experiments (single dots).

**Figure 5 biomedicines-10-02324-f005:**
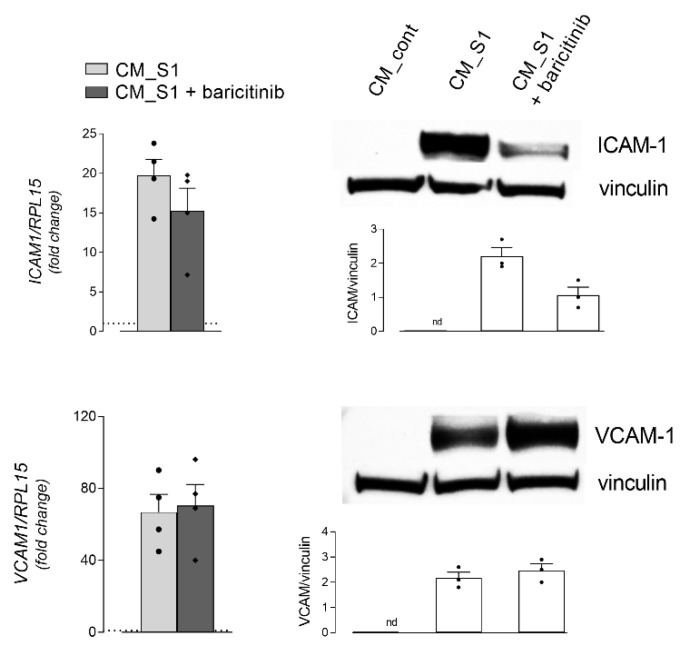
HUVEC were incubated in MDM-conditioned medium obtained by incubating MDM for 24 h in the absence (CM_cont) or in the presence of 5 nM S1 (CM_S1); baricitinib was added to CM_S1 1 h before the experiment and maintained during the whole experiment. After 4 h, the expression of the indicated genes (left panels) was measured by means of RT-qPCR and calculated relatively to CM_cont (=1; dotted line) upon normalization for the housekeeping gene *RPL15*, by employing the 2^∆Ct^ method. Bars are means ± SEM of four experiments (single dots), each performed in duplicate. After 6 h, the amount of the corresponding proteins (right panels) was assessed by means of Western Blot analysis (see Materials and Methods); representative blots are shown, along with the mean of the densitometric analyses (bars) of three different experiments (single dots).

**Figure 6 biomedicines-10-02324-f006:**
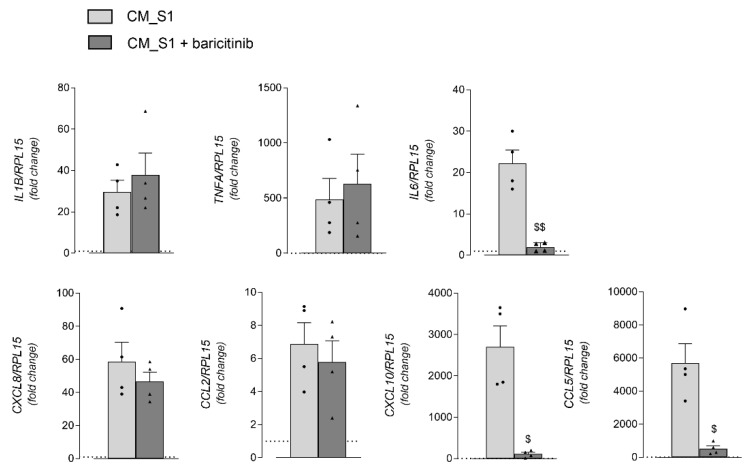
HUVECs were incubated for 4 h as described in Figure 5. The expression of the indicated genes was measured by means of RT-qPCR and calculated relatively to CM_cont (=1; dotted line) upon normalization for the housekeeping gene *RPL15*, by employing the 2^∆Ct^ method. Bars are means ± SEM of four experiments (single dots), each performed in duplicate. ^$^
*p* < 0.05, ^$$^
*p* < 0.01 vs. S1-treated cells with with Student’s *t* test.

**Table 1 biomedicines-10-02324-t001:** Sequence of primers pairs employed for RT-qPCR analysis.

Gene/Protein	Forward Primer	Reverse Primer
*RPL15*/RPL15	Hs03855120_g1 (TaqMan^®^ Assay, Thermo Fisher Scientific)	
*IL1B*/IL-1β	Hs99999029_m1 (TaqMan^®^ Assay, ThermoFisher Scientific)	
*IL6*/IL-6	AACCTGAACCTTCCAAAGATGG	TCTGGCTTGTTCCTCACTACT
*TNFA*/TNFα	ATGAGCACTGAAAGCATGATCC	GAGGGCTGATTAGAGAGAGGTC
*CXCL8*/IL-8	ACTGAGAGTGATTGAGAGTGGAC	AACCCTCTGCACCCAGTTTTC
*CXCL10*/IP-10	GTGGCATTCAAGGAGTACCTC	TGATGGCCTTCGATTCTGGATT
*CCL2*/MCP-1	CAGCCAGATGCAATCAATGCC	TGGAATCCTGAACCCACTTCT
*CCL3*/MIP-1α	AGTTCTCTGCATCACTTGCTG	CGGCTTCGCTTGGTTAGGAA
*CCL4*/MIP-1β	CTGTGCTGATCCCAGTGAATC	TCAGTTCAGTTCCAGGTCATACA
*CCL5*/RANTES	CTCCCCATATTCCTCGGACA	GTTGATGTACTCCCGAACCC
*ICAM1*/ICAM-1	TGAACCCCACAGTCACCTATG	CTCGTCCTCTGCGGTCAC
*VCAM1*/VCAM-1	GGGAAGATGGTCGTGATCCTT	TCTGGGGTGGTCTCGATTTTA

## Data Availability

Data are contained within the article.

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
