# Peer review of "The JAK1/2 Inhibitor Baricitinib Mitigates the Spike-Induced Inflammatory Response of Immune and Endothelial Cells In Vitro"

_biomedicines, 2022, doi:10.3390/biomedicines10092324_

Round 1

Reviewer 1 Report

The methods section is unacceptable. Especially the description of the RNA expression quantification is inaccurate and contains errors: example, Reference 24 does not give the required information.

"Hs03855120...(thermofisher actually gives LINC00333 for this code)" needs correction. A different code is valid for RPL15 (Hs06641691). The best would be to give the actual sequence (given by Thermofisher as part of intron 4).

Optional:

Several papers are missing from the introduction (or discussion, unless the paper explicitly places the presented work within a specific frame). Examples are:

https://www.ncbi.nlm.nih.gov/pmc/articles/PMC8420181/

https://www.ncbi.nlm.nih.gov/pmc/articles/PMC7270794/

https://www.ncbi.nlm.nih.gov/pmc/articles/PMC7300657/

JAK inhibition reduces SARS-CoV-2 liver infectivity and modulates inflammatory responses to reduce morbidity and mortality

In respect to the background, there are comparisons between baricitinib and tocilizumab, in several papers.

Some seemed to find similar clinical benefit, with the exception of geriatric patients.

In general, the subject of the importance of the topic, and the state of the art, needs some improvement, unless there is a very convincing statement on the novelty of the presented findings within another context.

Minor. Few expressions such as "huge" can be replaced by words such as "substantial", etc.

Author Response

We would like to thank the Reviewer for the criticisms he/she raised, that significantly helped improving our manuscript; according to his/her observations, we revised the manuscript as specified below.

Q1. The methods section is unacceptable. Especially the description of the RNA expression quantification is inaccurate and contains errors: example, Reference 24 does not give the required information.

A1. The section “Methods” has been improved, so as to cope with the Reviewer’s request

Q2. "Hs03855120...(thermofisher actually gives LINC00333 for this code)" needs correction. A different code is valid for RPL15 (Hs06641691). The best would be to give the actual sequence (given by Thermofisher as part of intron 4).

A2. We apologize, but we do not understand the issue raised by the Reviewer: the Assay ID He/She suggests (Hs06641691) actually targets RPL15, but is a test for the analysis of copy number variation (TaqMan® Copy Number Assays).

The primer pair we employed for the analysis of RPL15 expression is Hs03855120 (TaqMan® Gene Expression Assay), as already indicated in the original version. All the specifications of the assay are available when entering the Assay ID on the website of the supplier, as in the following link:

https://www.thermofisher.com/order/genome-database/?pearUXVerSuffix=pearUX2&elcanoForm=true#!/ge/assays/ge_all/?keyword=hs03855120&SID=srch-uc-gex-hs03855120&mode=and

Q3. Several papers are missing from the introduction (or discussion, unless the paper explicitly places the presented work within a specific frame). Examples are:

https://www.ncbi.nlm.nih.gov/pmc/articles/PMC8420181/

https://www.ncbi.nlm.nih.gov/pmc/articles/PMC7270794/

https://www.ncbi.nlm.nih.gov/pmc/articles/PMC7300657/

JAK inhibition reduces SARS-CoV-2 liver infectivity and modulates inflammatory responses to reduce morbidity and mortality

In respect to the background, there are comparisons between baricitinib and tocilizumab, in several papers.

Some seemed to find similar clinical benefit, with the exception of geriatric patients.

In general, the subject of the importance of the topic, and the state of the art, needs some improvement, unless there is a very convincing statement on the novelty of the presented findings within another context.

A3. The section Introduction has been improved and widened with new citations that actually better contextualize our findings; we thank the Reviewer for the useful advices.

Q4. Minor. Few expressions such as "huge" can be replaced by words such as "substantial", etc.

A4. The text has been revised according to the comment of the Reviewer.

Reviewer 2 Report

In this manuscript Barilli et al.  have studied the activity that the JAK-STAT inhibitor baricitinib exerts on the production of cytokines induced by the S1 protein from SARS-CoV-2 in human monocyte-derived macrophages (MDMs) and endothelial cells (HUVECs). They found that baricitinib decreased the production of IL6, TNF-a, IP-10 and MCP-1 in S1-treated MDMs, and that this inhibitor effectively reduced the production of IL-6, IP-10 and MCP-1 in HUVECs exposed to conditional medium from S1-treated MDMs. However, despite those actions, baricitinib changed neither neutrophil migration towards S1-containing conditional medium nor S1-induced expression of ICAMs in endothelial cells. Moreover, the authors show that S1 stimulates STAT and NF-kB phosphorylation/activation in MDMs and confirm that in these cells baricitinib selectively inhibits STAT activation, suggesting that the JAK-STAT signalling pathway may play an important role in the storm of cytokines associated with severe COVID-19, but that, taking into account the ineffective activity of baricitinib on neutrophil migration and expression of ICAMs in endothelial cells, the JAK-STAT inhibitor would be of limited efficacy for the treatment of COVID-19 at this stage of the disease.

Major comment

The work by Barilli et al. represents an interesting new approach to study the mechanisms by which the JAK-STAT signalling pathway is involved in the regulation of cytokine production in severe COVID-19 disease, and the possibility to use the JAK-STAT inhibitor baricitinib as a new treatment to control exacerbated host immune responses against SARS-CoV-2. Experimental approaches have been well performed and results are novel and interesting, although some of them require further clarification.

Some specific points

1.- Figure 3 and Line 208: The authors stated that “the presence of baricitinib in CM_S1….. promotes the migration of neutrophils to the same extent as CM_S1”. From this figure, and taking into account that no significant differences exist between CM_S1 and CM_S1+baricitinib, it seems that baricitinib does not promote cell migration. The authors could express this result more accurately.

2.- Figure 3: Cell migration needs to be studied in much more detail. Cell migration through matrix proteins (collagen/fibronectin) and endothelial cell monolayers should be performed before to conclude that baricitinib does not affect S1-dependent cell migration. On the other hand, an important function has been assigned to cells of the adaptive immune response in the cytokine storm of severe COVID-19. Does S1-induced conditional medium from MDMs affect T cell migration? Which would be the activity of baricitinib in S1-conditioned T cell migration?

3.- Figure 5: The action of S1-induced conditional medium from MDMs and baricitinib on VCAM-1 and ICAM-1 expression in HUVECs should be confirmed by flow cytometry. Flow cytometry analysis of the expression of ICAMs on the plasma membrane of endothelial cells may be a more valuable method to study their functionality than RT-qPCR or Western blot of total cell extracts.

4.- Regarding the inhibition of the expression of TNF-a by baricitinib, there is an apparent discrepancy between the cell types studied. The authors should convincingly discuss why baricitinib inhibits S1-induced TNF-a in MDMs but not in HUVECs  

Author Response

We would like to first thank the Reviewer for his/her collaborative criticisms; here below he/she can find the answers to his/her questions.

Q1. Figure 3 and Line 208: The authors stated that “the presence of baricitinib in CM_S1….. promotes the migration of neutrophils to the same extent as CM_S1”. From this figure, and taking into account that no significant differences exist between CM_S1 and CM_S1+baricitinib, it seems that baricitinib does not promote cell migration. The authors could express this result more accurately.

A1. We agree with the Reviewer that the sentence was confusing, so we rewrote it. We hope that this makes our finding better appreciable.

Q2. Figure 3: Cell migration needs to be studied in much more detail. Cell migration through matrix proteins (collagen/fibronectin) and endothelial cell monolayers should be performed before to conclude that baricitinib does not affect S1-dependent cell migration. On the other hand, an important function has been assigned to cells of the adaptive immune response in the cytokine storm of severe COVID-19. Does S1-induced conditional medium from MDMs affect T cell migration? Which would be the activity of baricitinib in S1-conditioned T cell migration?

A2. According to the Reviewer’s suggestions, the transendothelial migration of neutrophils toward the conditioned medium has been evaluated, along with that of human T lymphocytes. New findings are now shown in the revised version of the ms.

Q3. Figure 5: The action of S1-induced conditional medium from MDMs and baricitinib on VCAM-1 and ICAM-1 expression in HUVECs should be confirmed by flow cytometry. Flow cytometry analysis of the expression of ICAMs on the plasma membrane of endothelial cells may be a more valuable method to study their functionality than RT-qPCR or Western blot of total cell extracts.

A3. The induction of ICAM and VCAM by different stimuli is mostly regulated at transcriptional level;  thus, an increased expression of these adhesion molecules is a well recognized hallmark of endothelial activation. In our study, we observed a 20- and 60-fold induction of ICAM and VCAM at mRNA level, while the proteins, completely absent in controls, were impressively induced in stimulated cells. We think, therefore, that these results clearly demonstrate the activation of endothelial cells under our conditions and that flow cytometry analysis would provide no additional information.

Q4. Regarding the inhibition of the expression of TNF-a by baricitinib, there is an apparent discrepancy between the cell types studied. The authors should convincingly discuss why baricitinib inhibits S1-induced TNF-a in MDMs but not in HUVECs  

A4. The discrepancy evidenced by the Reviewer in the effects of baricitinib on the expression of TNFa is due to the different experimental conditions adopted for MDMs and HUVEC: while MDMs were incubated with spike S1 protein, endothelial cells were treated with conditioned medium from S1-activated MDM. Given the different effects of baricitinib, it is likely to suppose that the induction of TNFa is mediated by different molecular pathways under the two conditions. This issue, as suggested by Reviewer, is now addressed in Discussion.

Round 2

Reviewer 1 Report

Accept in present form

Reviewer 2 Report

The authors have addressed all my concerns. The manuscript is now 
sufficiently improved to publication in Biomedicines.